

# Swarm v2: highly-scalable and high-resolution amplicon clustering

Frédéric Mahé[1], Torbjørn Rognes[2,3], Christopher Quince[4], Colomban de Vargas[5,6] and Micah Dunthorn[1]

[1] Department of Ecology, Technische Universität Kaiserslautern, Kaiserslautern, Germany
[2] Department of Informatics, University of Oslo, Oslo, Norway
[3] Department of Microbiology, Oslo University Hospital, Rikshospitalet, Oslo, Norway
[4] Warwick Medical School, University of Warwick, Warwick, United Kingdom
[5] UMR 7144, EPEP–Évolution des Protistes et des Écosystèmes Pélagiques, Station Biologique de Roscoff, CNRS, Roscoff, France
[6] UMR7144 Station Biologique de Roscoff, Sorbonne Universités, UPMC Univ Paris 06, Roscoff, France

## ABSTRACT

Previously we presented Swarm v1, a novel and open source amplicon clustering program that produced fine-scale molecular operational taxonomic units (OTUs), free of arbitrary global clustering thresholds and input-order dependency. Swarm v1 worked with an initial phase that used iterative single-linkage with a local clustering threshold ($d$), followed by a phase that used the internal abundance structures of clusters to break chained OTUs. Here we present Swarm v2, which has two important novel features: (1) a new algorithm for $d = 1$ that allows the computation time of the program to scale linearly with increasing amounts of data; and (2) the new fastidious option that reduces under-grouping by grafting low abundant OTUs (e.g., singletons and doubletons) onto larger ones. Swarm v2 also directly integrates the clustering and breaking phases, dereplicates sequencing reads with $d = 0$, outputs OTU representatives in fasta format, and plots individual OTUs as two-dimensional networks.

## INTRODUCTION

Traditional *de novo* amplicon clustering methods that can handle large high-throughput sequencing datasets (e.g., *Edgar, 2010*; *Ghodsi, Liu & Pop, 2011*; *Fu et al., 2012*) suffer from two fundamental problems. First, they rely on an arbitrary fixed global clustering threshold to group amplicons into molecular operational taxonomic units (OTUs). Global clustering thresholds have rarely been justified and are not applicable to all taxa and marker lengths (e.g., *Caron et al., 2009*; *Nebel et al., 2011*; *Dunthorn et al., 2012*; *Brown et al., 2015*). Second, there is variability in the clustering results due to amplicon input order (*Koeppel & Wu, 2013*; *Mahé et al., 2014*).

To solve these problems, we previously introduced the open source Swarm v1 program that implemented an initial clustering phase written in C++, then a breaking phase written in Python (*Mahé et al., 2014*). Swarm's clustering phase (Fig. 1A) was novel in its approach to single linkage clustering in that, instead of using a global clustering (e.g., *Hartmann et al., 2012*; *Huse et al., 2010*), amplicons were iteratively added together using a

Corresponding authors
Frédéric Mahé,
mahe@rhrk.uni-kl.de
Torbjørn Rognes,
torognes@ifi.uio.no

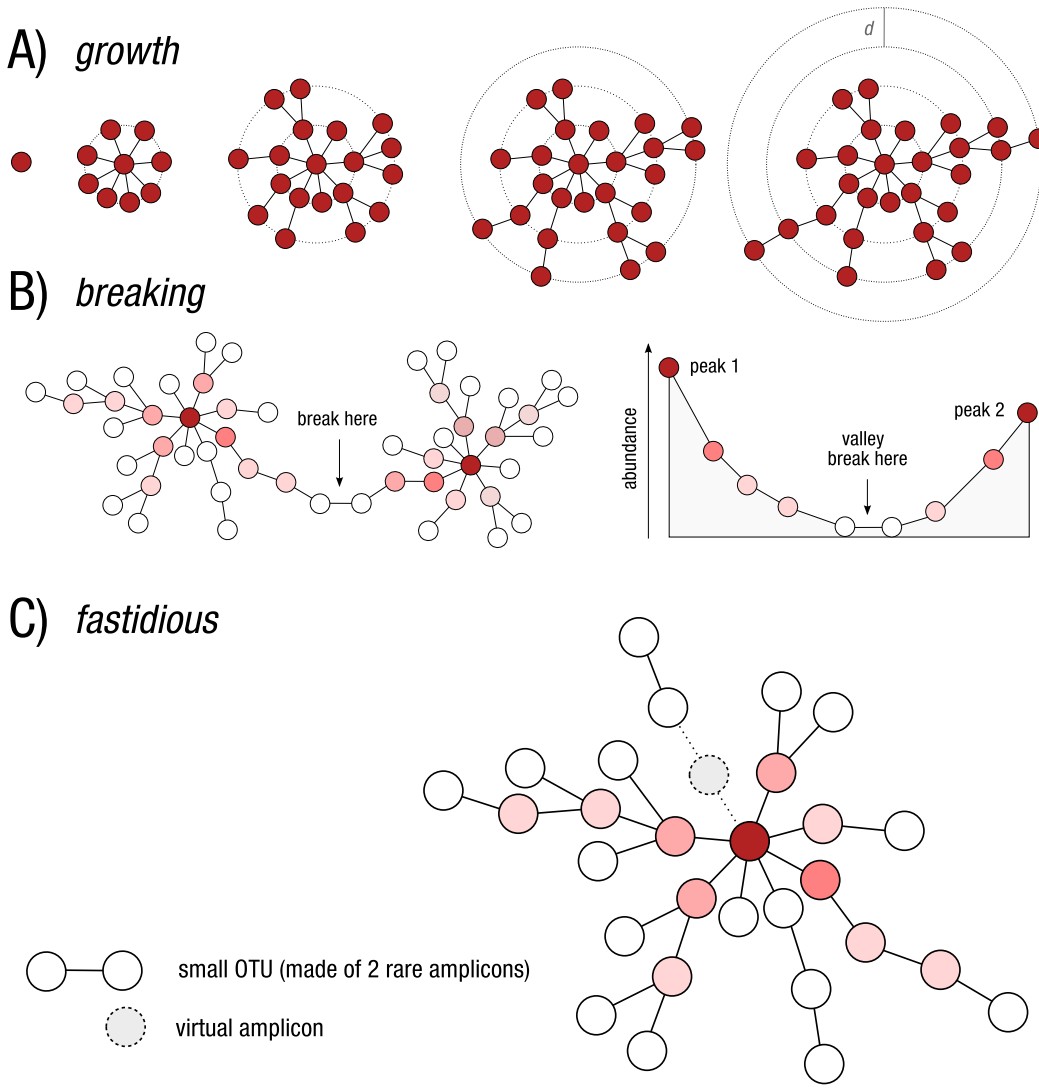

**Figure 1 Schematic view of Swarm's clustering and refinement approach.** (A) Swarm clusters amplicons iteratively by using a small user-chosen local threshold, *d*, allowing OTUs to grow to their natural limits, where no other amplicons can be added. (B) Swarm takes into account the abundance of each amplicon to produce higher resolution clusters, by not allowing the formation of amplicon chains. The darker the red, the higher the abundance. (C) The fastidious option avoids under-grouping (e.g., the production of small OTUs such as singletons and doubletons) by postulating the existence of virtual linking amplicons to graft smaller OTUs onto larger ones.

small local clustering threshold ($d$) until no more amplicons could be added. Using $d = 1$ produced the highest resolution OTUs. Swarm's breaking phase (Fig. 1B) was novel in that it used the abundance of amplicons to reveal the internal structure of potentially chained OTUs (i.e., a low abundant link between high abundant amplicons). These chained OTUs were then refined by splitting them.

Since its introduction, Swarm v1 has been used in a variety of datasets (*De Vargas et al., 2015*; *Filker et al., 2015*; *Lima-Mendez et al., 2015*; *Mahé et al., 2015*; *Oikonomou et al., 2015*). However, since the breaking phase was written in Python, it lacked scalability and

was cumbersome to use. Kopylova et al. (pers. comm., 2015) also found that in comparison to other clustering methods, Swarm v1 tended to produce relatively more low abundant OTUs; e.g., singletons and doubletons. And most importantly, Swarm v1 and other current *de novo* algorithms could not cluster today's largest high-throughout sequencing datasets within a reasonable amount of time (*Rideout et al., 2014*). Here we introduce Swarm v2 to help solve these problems, as well as to introduce new and useful features.

## MATERIAL AND METHODS

### Linear complexity *de novo* clustering approach

Today's largest amplicon datasets contain hundreds of millions of amplicons and pose a computational challenge to *de novo* clustering methods. Because of this scalability problem, *Rideout et al. (2014)* proposed using a mixed clustering approach with an initial closed-reference clustering that compares the amplicons to what is known in taxonomic reference databases to capture most of the data, followed by a *de novo* clustering with the remaining amplicons. We feel that using only *de novo* clustering is the most powerful approach when working with amplicons from unexplored environments that lack sufficient taxonomic reference databases, or with rare taxa that were previously missed in already-sampled environments. We therefore worked to improve Swarm's scalability.

Like other current *de novo* clustering approaches, Swarm v1 presented an apparent quadratic behavior in that it needs to perform a number of comparisons that grows as the square of the number of amplicons. In Swarm v2, we first reduced computational time by improving the multithreading and making a better usage of multi-core CPUs. We further reduced computational time by using a novel algorithmic approach. This linear complexity approach only applies for $d = 1$, which is Swarm's default and preferred parameter as it produces the highest resolution clusters.

As background to this linear approach, let us consider a nucleic sequence $S$ made of As, Cs, Gs and Ts. A "microvariant" is a sequence with one difference ($d = 1$) to the original sequence $S$. How many distinct microvariants can derive from $S$? In a sequence $S$ of length $L$, each position can be substituted with 3 different bases, so there are $3L$ possible microvariants due to substitutions. Each position in $S$ can be deleted once, so there are $L$ possible microvariants due to deletions. Insertions are more complicated. An insertion can happen before or after each position in the sequence $S$, and four different nucleotides can be inserted resulting in $4(L + 1)$ microvariants. However, some insertions will result in the same microvariant: for example, inserting a "G" before or after a "G" will result in the same sequence "GG." As that situation arises for all positions in $S$ but one, the maximum number of distinct insertions is not $4(L + 1)$, but $3(L + 1) + 1 = 3L + 4$. In total, the maximum number of microvariants that can be obtained from a given sequence $S$ of length $L$ is $3L + L + 3L + 4 = 7L + 4$.

As stated above, different sequence modifications can produce the same microvariant. The final number of distinct microvariants depends on the number of homopolymer stretches in the sequence. In the extreme situation where the sequence is entirely made of one type of nucleotide, the number of microvariants due to deletions drops from $L$ to 1. For example, if $S$ is entirely made of "G," all possible deletions yield the same

microvariant. The total number of distinct microvariants then drops to its minimum value: $3L + 1 + 3(L + 1) + 1 = 6L + 5$.

The number of distinct microvariants that can be obtained from a sequence $S$ of length $L$ then varies from $6L + 5$ to $7L + 4$. In practice, it means that a typical high-throughput sequencing 16S rRNA sequence of 130 nucleotides will yield at least 785 microvariants and at most 914, and that the number of microvariants will increase linearly with the sequence length. With current sequencing technologies read length increases at a slower rate than read number, and is safe to assume it will continue to do so in the foreseeable future.

Based on these characteristics of microvariants, we switched from an approximate-string comparison approach to an exact-string comparison approach. That is, for a given amplicon, instead of doing an exact pairwise alignment comparison against all available amplicons in the pool that have yet to be subsumed into an OTU, Swarm v2 generates all possible microvariants of the amplicon and checks whether or not they are present in the amplicon pool using a hash table. This exact-string search approach is extremely fast, and does not depend on the number of available amplicons in the pool. Therefore, the apparent computational complexity changes from $n^2$ to $n \times L$, where $L$ is the average amplicon length.

## Reducing under-grouping

As observed by Kopylova et al. (pers. comm., 2015), Swarm v1 tended to produce relatively more low abundant OTUs; e.g., singletons and doubletons. This problem is due to Swarm's approach that depends on the existence of a continuous path of linked amplicons. Linking amplicons can be missing, especially when sequencing is shallow. When this occurs, there can be under-grouping of closely related amplicons leading to small OTUs surrounding a larger OTU.

To address this problem in Swarm v2, we introduced a new step—called the fastidious option—to graft low abundant OTUs onto more abundant ones by postulating a linking amplicon (Fig. 1C). A low abundant OTU is defined as an OTU with a total abundance lower than 3, i.e., an abundance of one (singletons) or two (doubletons). That default threshold value can be modified by users with the option -b. In practice, microvariants (independent of the microvariants produced in the clustering phase) are produced for all the amplicons belonging to low abundant OTUs and stored in a Bloom filter (a probabilistic dictionary). Microvariants are then produced for high-abundant amplicons and cross-checked against the Bloom filter. The fastidious option can consume a large amount of memory, but is apparently linear in terms of computation time (see Results). The user does have control over memory usage and can exchange memory space for computation time. As of now, the fastidious option can only be used with $d = 1$, which is Swarm's default and recommended $d$ value. With higher $d$ values, the time and space complexity of our fastidious algorithm grows too fast to be practical.

The fastidious option can be viewed as a way to reduce data loss, as many researchers conservatively consider low abundant OTUs as spurious errors and remove them from downstream analyses (*Behnke et al., 2011*; *Kunin et al., 2010*). With the fastidious option, though, one can retain many of these amplicons by attaching them to more

abundant OTUs. In contrast with an increase of $d$, the fastidious option does not slow down computation and does not degrade the clustering resolution; i.e., it reduces the under-grouping of amplicons without inducing much over-grouping (see section "Statistics on mock-communities" in File S1).

## Other new and useful features

In Swarm v2 we introduce a number of options improving both speed and usability. First, there is a simpler user command line interface. For example, the breaking phase is now written in C++ and is performed directly during the growth phase, which further significantly reduces computation time. We chose to implement a strict, simple, non-parametric breaking model that prevents any increase in abundance along a continuous amplicon path (Fig. 1B). Breaking of linked chains can be deactivated.

Second, Swarm v2 extends the notion of clustering by allowing the option $d = 0$. Users can now dereplicate their sequencing reads into strictly identical amplicons (sensu *Mahé et al., 2015*; i.e., reads that have exactly the same sequences with no substitutions, insertions, or deletions). This fast dereplication approach uses the same algorithm as in VSEARCH (https://github.com/torognes/vsearch).

Third, Swarm v2 can output OTU representative amplicons in fasta format. A representative is the most abundant amplicon of an OTU, and its abundance is updated to reflect the total OTU abundance. OTU representatives are normally used for downstream community-comparative, novel-diversity, and ecosystem-functioning questions.

Fourth, Swarm v2 offers the possibility to visualize the internal structure of OTUs, which allows the user to gain further knowledge of its underlying genetic and ecological diversity (Figs. 2 and 3). These plots are in the form of a network projected in two-dimensions. Edges in these networks only represent the parameter $d$ used; the length of the edges carries no information. The nodes in the networks represent amplicons. The abundance information of these amplicons is represented in three ways: the size of the node, the color of the node, and text when its abundance value is 10 or more.

## Analyses

To demonstrate the apparent linear complexity of Swarm v2, we analyzed 16S rRNA reads from the Earth Microbiome Project (*Gilbert, Jansson & Knight, 2014*), which is the largest amplicon dataset currently available. The following swarm commands were used: `swarm -d 1 in.fasta`, and `swarm -d 1 -f in.fasta`. To illustrate over- and under-grouping of amplicons, the importance of the breaking phase, high-resolution clustering, and Swarm's ability to visualize OTUs' internal structures, we used 18S rRNA amplicon data from the BioMarKs consortium (*Logares et al., 2014*) that sampled European near-shore marine sites. The PR2 v203 reference database was used for taxonomic assignment (*Guillou et al., 2013*). The full methods can be found online in html format (File S1).

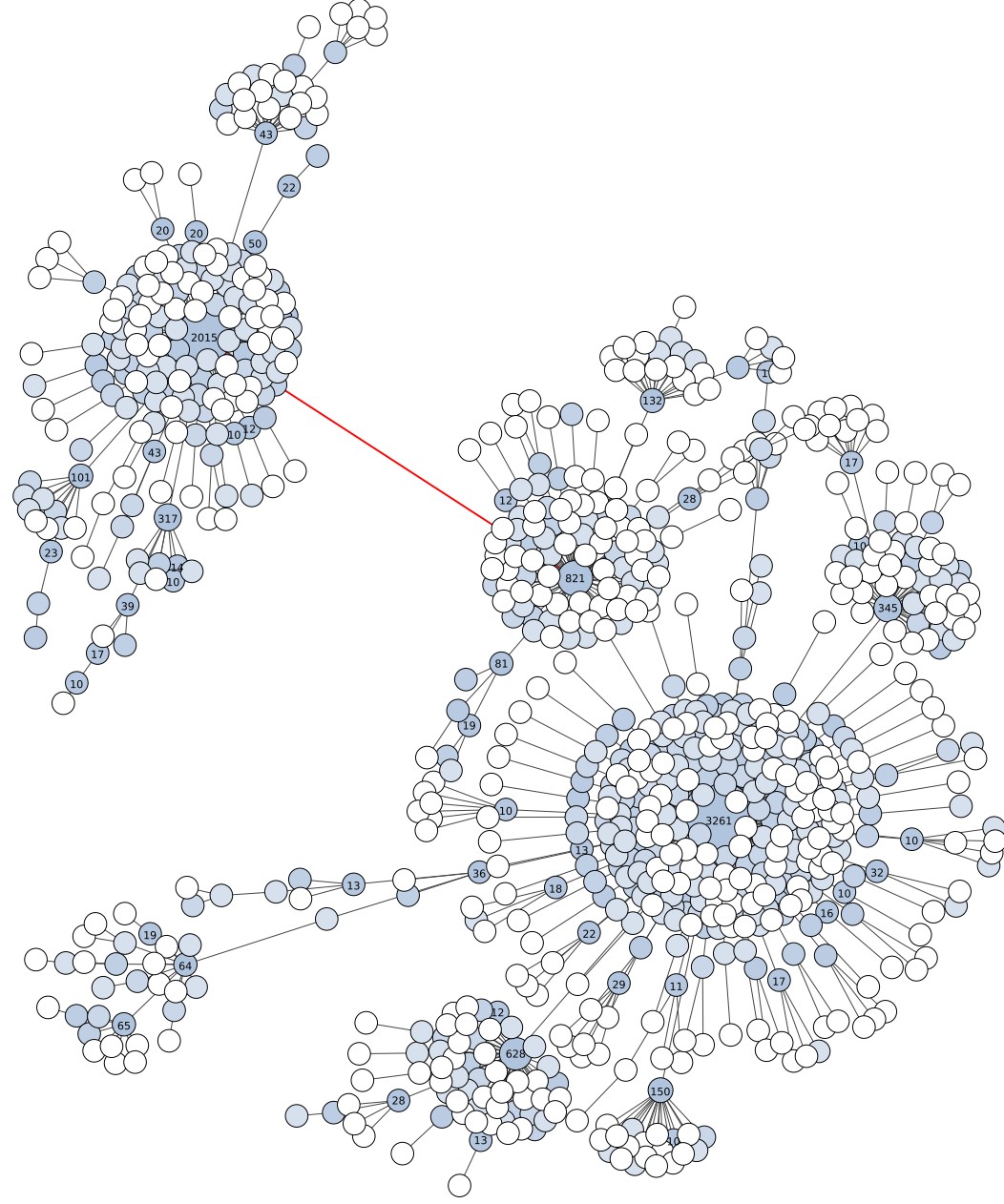

**Figure 2 Graphical representation of an OTU produced by Swarm (breaking and grafting phases deactivated) when clustering the BioMarKs 18S rRNA V9 dataset (amplicons are appr. 129 bp in length).** Nodes represent amplicons. Node size, color and text annotations represent the abundance of each amplicon. Edges represent one difference (substitution, deletion or insertion); the length of the edges carries no information. The red-colored edge indicates where Swarm's breaking phase cuts when it is not deactivated, resulting into two high abundant OTUs, each being assigned to a different genus of Collodaria (Radiolaria).

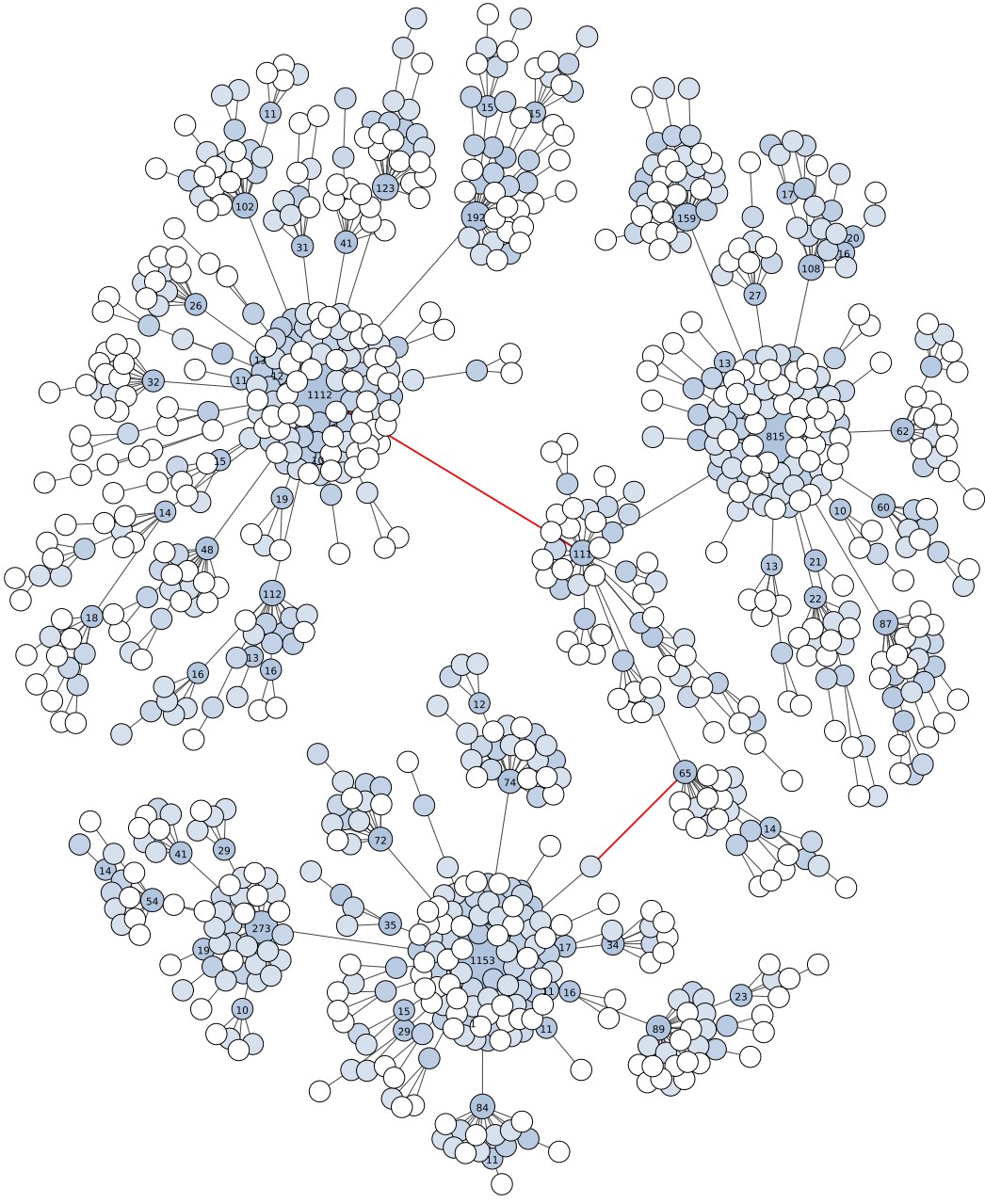

**Figure 3 Graphical representation of an OTU produced by Swarm (breaking and grafting phases deactivated) when clustering the BioMarKs 18S rRNA V4 dataset (amplicons are appr. 380 bp in length).** Nodes represent amplicons. Node size, color and text annotations represent the abundance of each amplicon. Edges represent one difference (substitution, deletion or insertion); the length of the edges carries no information. The red-colored edges indicate where Swarm's breaking phase cuts when it is not deactivated, resulting into three high abundant OTUs, each being assigned to a different taxa of Cnidaria (Metazoa).

## RESULTS AND DISCUSSION

### Time and space benchmarks

For $d = 1$, Swarm's default parameter, using the full- and sub-datasets of the Earth Microbiome Project we were able to evaluate Swarm v2's clustering time and memory usage. These timing experiments were obtained with Swarm v2.1.1 on a machine with 1,024 GB of RAM running Red Hat CentOS v6.6 and Linux kernel v3.9.1 on four Intel Xeon E5-4620 processors (2.2 GHz) having a total of 32 physical cores. Swarm was run with 8 threads (option "-t 8"), breaking activated (default behaviour), and memory limited to 240 GB ("-c 245760"). The times indicated below are the averages of four runs. With the sub-dataset of 154,896,650 strictly identical amplicons (representing 1,277,640,415 reads), Swarm without the fastidious option took 1 h and 45 min $\pm$ 1 min. With the full-dataset of 314,871,149 strictly identical amplicons (representing 2,254,207,945 reads), Swarm without the fastidious option took 3 h and 41 min $\pm$ 1 min. Doubling of dataset size approximately doubles the run time, confirming the apparent linear time complexity. Therefore, if the size of the Earth Microbiome Project were to increase ten times, it should take about ten times longer to cluster it (less than two days). These fast times of Swarm v2 contrast with the estimated computational time of UCLUST v6.1 as inferred by *Rideout et al. (2014)*. Using a smaller partial-dataset of the Earth Microbiome Project consisting of only 660,000,000 reads (that dereplicate into a unspecified number of strictly identical amplicons), *Rideout et al. (2014)* estimated UCLUST's runtime to 150 days on a 8-core computer.

With the sub-dataset representing 24 GB of input data, the memory usage of Swarm v2 with $d = 1$ was 41 GB. With the full-dataset representing 49 GB of input data, the memory usage was 86 GB. Memory requirements can therefore be estimated to be approximately equal to the size of the input dataset plus 2/3.

When clustering at $d = 1$ and using the fastidious option, the total computational time of the sub-dataset was 4 h and 59 min $\pm$ 1 min, which resulted in 40.0% fewer OTUs in total. The total computational time of the full-dataset took 11 h and 28 min $\pm$ 5 min, which resulted in 38.3% fewer OTUs in total. This considerable reduction in the number of singletons and doubletons in both datasets helps solve the problem found by Kopylova et al. (pers. comm., 2015). The computation time is about three times longer when using the fastidious option than without it.

The total memory usage of $d = 1$ with the fastidious option for the sub-dataset was 114 GB, while it was 239 GB (capped) for the full-dataset. This amount of memory might not be available to all users. Therefore we have implemented two options to control and cap memory usage of the fastidious option: by defining the maximum memory footprint, and by adjusting the size of the Bloom filter entries. Both of these options allow users to trade computational time for memory space.

### OTU visualizations

We provide examples of Swarm v2's graphical representation of the internal structure of its high-resolution OTUs by using V4 and V9 18S rRNA amplicons. In both cases the breaking phase and fastidious option were turned off. With the V9 data (about 129 bp in length), the graph shows two high abundant OTUs linked by one lower abundant amplicon (Fig. 2).

The number of nucleotide differences between these two linked OTUs is only two, or about 98.4% similarity. If the breaking phase and fastidious option were applied to these V9 amplicons, nine separate OTUs would have been formed: two high abundant, and seven low abundant. These two high abundant OTUs are taxonomically assigned to different genera of Collodaria (Radiolaria). On the same V9 amplicons, UCLUST v6 (as well as v7 and v8) using a global clustering threshold of 97% similarity produced 37 OTUs (one high abundant, and 36 low abundant). The one high abundant OTU from UCLUST lumped the two Collodaria genera, thus masking meaningful biological data.

With the V4 amplicons (about 380 bp in length), the graph shows three high abundant OTUs linked by one to three low abundant amplicons (Fig. 3). The number of nucleotide differences between these three linked OTUs is only two and four, or about at least 98.9% similarity. If the breaking phase and fastidious option were applied to these V4 amplicons, seven separate OTUs would have been formed: three high abundant, and four low abundant. These three high abundant OTUs are assigned to different taxa of Cnidaria. On the same V4 amplicons, UCLUST v6 (as well as v7 and v8) produced only one OTU with the widely used global clustering threshold of 97% similarity, again masking meaningful biological data.

These amplicon data show that, compared to UCLUST, Swarm v2 can distinguish higher-resolution clusters and reduces both over-grouping and under-grouping on a range of marker lengths. In both of these amplicon examples, Swarm v2 is able to distinguish different taxa, while UCLUST conceals them.

## Outlook

We are currently working on a number of fronts to continue making Swarm harder, better, faster, stronger. For example, preliminary experiments indicate that with a novel multi-threading approach for $d \geq 2$ a ten-fold increase in speed could be obtained (although $d \geq 2$ will still be quadratic in behavior). Internally encoding nucleotides on two bits instead of eight bits may help reduce both memory consumption and computational time. Additional computation time can be saved by merging the fastidious option with the initial clustering phase. To facilitate its usage, Swarm v2 can be included in QIIME (*Caporaso et al., 2010*), which already offers Swarm v1.2, and in Galaxy (*Goecks, Nekrutenko & Taylor, 2010*).

In summary, Swarm v2 is a highly-scalable approach that uses a local clustering threshold to produce high-resolution *de novo* OTUs and reduces low abundant OTUs. Swarm v2 is an optimized C++ program able to handle many hundreds of millions of amplicons. It is open source and freely available at https://github.com/torognes/swarm under the GNU Affero General Public License version 3.

## ACKNOWLEDGEMENTS

We would like to thank the Earth Microbiome Project for the use of their data and their constructive comments. Daft Punk provided the background music. We are grateful to the computational resources at the Regional Computing Center at the University of Kaiserslautern, and the Abel computing cluster at the University of Oslo.

### Funding

FM and MD were supported by the Deutsche Forschungsgemeinschaft (grant #DU1319/1-1). CQ is funded by an EPSRC Career Acceleration Fellowship—EP/H003851/1. CdeV were supported by the EU EraNet BiodivErsA program BioMarKs (grant #2008-6530) and the French government "Investissements d'Avenir" project OCEANOMICS (ANR-11-BTBR-0008) and the EU FP7 program MicroB3 (contract number 287589). The funders had no role in study design, data collection and analysis, decision to publish, or preparation of the manuscript.

### Grant Disclosures

The following grant information was disclosed by the authors:
Deutsche Forschungsgemeinschaft: #DU1319/1-1.
EPSRC Career Acceleration Fellowship: EP/H003851/1.
EU EraNet BiodivErsA program BioMarKs: #2008-6530.
French government "Investissements d'Avenir" project OCEANOMICS: ANR-11-BTBR-0008.
EU FP7 program MicroB3: 287589.

### Competing Interests

The authors declare there are no competing interests.

### Author Contributions

- Frédéric Mahé and Torbjørn Rognes conceived and designed the experiments, performed the experiments, analyzed the data, wrote the paper, prepared figures and/or tables, reviewed drafts of the paper.
- Christopher Quince and Colomban de Vargas conceived and designed the experiments, contributed reagents/materials/analysis tools, reviewed drafts of the paper.
- Micah Dunthorn conceived and designed the experiments, analyzed the data, wrote the paper, prepared figures and/or tables, reviewed drafts of the paper.

### Data Availability

https://github.com/torognes/swarm.

### Supplemental Information

Supplemental information for this article can be found online at http://dx.doi.org/10.7717/peerj.1420#supplemental-information.

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
