# Peer review of "Swarm v2: highly-scalable and high-resolution amplicon clustering"

_PeerJ, doi:10.7717/peerj.1420_

## Round 0.1 · original submission · Major Revisions

As you will see, both referees are supportive of the work. However, they also have a number of good points for your consideration. Referee #1 notes that better and biologically relevant comparison between swarm and uclust would be useful for the reader. The reviewers also note several other points that will help to improve your manuscript.

Reviewer 1 ·

Basic reporting

No Comments

Experimental design

I wonder that whether fastidious option can only be applied when d=1, or it’s also available when d≥1. If I understand correctly, can we subsume a low abundance OTU into one OTU when the local abundance threshold between the low abundance OTU and the farthest layer is not greater than 2d, since the max value of local abundance threshold from both sides to the virtual linking amplicon is 1d. It is mentioned in the result that fastidious can reduce the total number of OTU, but I wonder if there is any means we can tell those OTUs which has been subsumed and the high abundance OTU are under the same taxon. I don’t know if the OTU reduction would lead to some information loss in those rare species, which might be concerned in research.
I am interested to see the results of the computational time, memory usage and the total number of OTUs when the breaking phase option is applied. I wonder if the seed choosing and the amplicon input and clustering order would impact on the final results.

Validity of the findings

No Comments

Additional comments

This paper describes some improvements for Swarm, a clustering program which does not influence by the input order or arbitrary global clustering thresholds. There are two major improvements mentioned. By improving the algorithm, computational time is greatly reduced. And settled the under-grouping by postulating the virtual linking amplicons.
Moreover, the program had been intergraded written in C++. Users can export the sequences in fasta format, when d, local clustering thresholds, =0. The internal structure of OTUs can be visualized in two-dimension plot.
Single linkage + break is a smart idea for clustering, but we do have some concerns about the original algorithm: using the number of differing bases (d=1 means one base difference) is also arbitrary like using distance threshold (not to mention that 97% is at least roughly corresponding to species level); sequences in a single linkage OTU are not affected by the seed chosen and clustering sequence, but the structure of the single linkage OTU will be affected by them (how these sequences are connected) and hence affecting the break process. From figure 2 and 3, we do see that some sequences can be linked into a single OTU even if the distance is relatively high between them (the same problem as single linakge clustering), can the author give a summary about the OTU size (for example, the average/highest distance between sequences and their centroid) for single linkage (or swarm without break), swarm (with break) and uclust so that we can understand how break can improve based on single linkage and not too stringent like uclust? Nevertheless, since this study is a refinement based on older software, we do not argue that the authors have to answer these.
Regarding this study, we are not sure why using different sized data to compare swarm and uclust, even when the speed is clearly much faster using swarm than using uclust; and how about giving some biological comparison between swarm and uclust using, for example, alpha and beta diversity so that the advantage is clearer rather than just theoretically better.

·

Basic reporting

No comments

Experimental design

In the result section they show that they are able to analyse the EMP dataset in a reasonable time on a standard server. It would be nice to add a comparison with SortMeRNA scince that is the default algorithm in Qiita, the storage and analysis platform for EMP.

To illustrate the effect of under-grouping with the fastidious option they use a 18S dataset. I think this section can be rephrased so that the main message, Swarm v2 reduces both over- and under-grouping, becomes more clear. For example, the results of the first and second version of Swarm can be compared. In the Swarm v1 paper a comparison with Usearch using a mock dataset is performed. A similar analysis could be useful here. Interesting to know is how the total numbers of OTUs differ for Swarm v2 and the cluster_smallmem / cluster_otus algorithms in Usearch on a mock dataset.

Validity of the findings

No comments

Additional comments

The authors first introduce the general concept of amplicon clustering applied in Swarm and how it differentiates from other clustering algorithms. Secondly they make clear that they made several improvements to the software. They clearly state why and how they changed the algorithm to scale linearly so it can handle large datasets. Also they introduced a new option (fastidious) to prevent the formation of low abundant OTUs.

The procedure to dereplicate and hash the sequences as described in the supplementary html file is different from the documentation on Github. The latter could be updated where an example using VSEARCH is given together with the Python hashing script. This script needs to be added to the Github repository. The Swarm command in the Supplementary html file is missing the '-z' flag so that it accepts the output from VSEARCH which contains abundance information in usearch style.

Furthermore Swarm could be improved by making it easier to produce a OTU table with counts for multiple samples, or even have this formatted in BIOM format. Currently there is an example awk script on the wiki (https://github.com/torognes/swarm/wiki/Working%20with%20several%20samples#produce-a-contingency-table-for-otus) which is not present in the scripts folder in the repository. This scripts needs to be updated as well since it does not work with abundance annotation in usearch style.

---

## Round 0.2 · accepted · Accept

The changes that have been made appropriately address the queries from the referees.